# Leukocyte Count Predicts Carotid Artery Stenosis in Men with Ischemic Stroke: Sub Study of the Preventive Antibiotics in Stroke Study (PASS)

**DOI:** 10.3390/jcm11247286

**Published:** 2022-12-08

**Authors:** Twan J. van Velzen, Jeffrey Stolp, Willeke F. Westendorp, Yvo B. W. E. M. Roos, Diederik van de Beek, Paul J. Nederkoorn

**Affiliations:** Department of Neurology, Amsterdam UMC, Location AMC, Amsterdam Neuroscience, Meibergdreef 9, 1105 AZ Amsterdam, The Netherlands

**Keywords:** atherosclerosis, ischemic stroke, internal carotid artery stenosis, leukocytes, thrombocytes

## Abstract

*Background*: Inflammation is important in the development of atherosclerosis. Research suggested sex-dependent differences for the value of inflammatory markers for risk stratification of stroke patients with internal carotid artery stenosis (ICAS). We investigated whether leukocytes and thrombocytes were associated with ≥50% ICAS in acute stroke and whether this was sex-dependent. Patients included in the Preventive Antibiotics in Stroke Study (PASS) were used. PASS is a randomized controlled trial that randomized between four days of preventive ceftriaxone intravenously or standard stroke care alone. It investigated whether ceftriaxone could improve functional outcome at three months after stroke. *Methods*: Patients included in PASS were evaluated for the predictive value of leukocytes and thrombocytes for ICAS. Ischemic stroke and TIA patients were selected out of PASS patients. Logistic regression analysis was performed adjusting for NIHSS and other covariates. Results: 2550 patients were included in PASS. 1413 of 2550 patients (55%) were evaluated in this sub study. Female patients showed a mean of 8.55 × 10^9^/L for leukocytes and 259 × 10^9^/L for thrombocytes. Men showed a mean of 8.29 × 10^9^/L for leukocytes and 224 × 10^9^/L for thrombocytes. Multivariate logistic regression analysis showed that leukocytes were independently associated with ICAS ≥ 50% in male patients (OR 1.094, *p* = 0.008), but not in female patients (OR 1.041, *p* = 0.360). Thrombocytes were not associated with ICAS. Conclusions: We conclude that blood leukocyte count independently predicts ICAS in men after acute stroke, but not in women. Clinical Trial unique identifier: ISRCTN66140176.

## 1. Introduction

Inflammation is a major driving force in initiation and progression of atherosclerosis [1]. Atherosclerosis can lead to internal carotid artery stenosis (ICAS), which is an important risk factor for stroke and stroke recurrence [2,3]. Immunomodulating therapies have emerged as potential treatment for atherosclerosis, but these therapies have not yet been established as beneficial in ICAS [4,5]. The population-based Tromsø study described an independent association between blood leukocyte count and carotid plaque area in men but not in women, when corrected for age, smoking, cholesterol, cardiovascular blood pressure, diabetes mellitus and the use of lipid lowering drugs [6]. To our understanding this association between blood leukocyte count and carotid plaque has not be studied in the acute stroke setting yet. The Tromsø study was a population based prospective study in which multiple health surveys were done. The role of the association between blood leukocyte count should therefore be considered in primary prevention. If an association is found in the acute stroke setting the consequences can be multiple. A role in the acute treatment or as secondary prevention could be discussed.

Differences between sexes are frequently observed for atherosclerotic disease but the cause is unknown [7]. Assessing sex-specific mechanisms in atherosclerosis may lead to a further understanding of the pathological mechanisms underlying ICAS. We aimed to assess the sex-dependent association between leukocyte count and ICAS in the acute stroke setting using the database of the Preventive Antibiotics in Stroke Study (PASS), a multicentre prospective randomized, open label, masked endpoint trial [8]. PASS investigated whether four days of ceftriaxone intravenously could improve functional outcome at three months after stroke compared to stroke care alone.

## 2. Materials and Methods

### 2.1. Study Population

PASS was conducted in the Netherlands, in which patients were assigned to either ceftriaxone intravenously for four days in combination with standard stroke care or standard stroke care alone (registration number ISRCTN66140176, date of first registration 6 April 2010). Patients with a stroke (haemorrhagic stroke, ischemic stroke or TIA) with an onset of symptoms less than 24 h and a National Institutes of Health Stroke Scale (NIHSS) ≥1, that were admitted to the hospital were included. The exclusion criteria were signs of infection during admission that required treatment with antibiotics, use of antimicrobials <24 h before admission, pregnancy, cephalosporin hypersensitivity, history of anaphylaxis from penicillin derivates, subarachnoid haemorrhage or imminent death [8,9]. Written informed consent from each patient or representative, and ethics approval by the Dutch ethics committee on research on humans were obtained. The study protocol conformed to the ethical guidelines of the 1975 Declaration of Helsinki.

### 2.2. Baseline Characteristics

Baseline patient characteristics were collected during the PASS, including age, sex, type of stroke, medication use and medical history (hypercholesterolemia, hypertension, diabetes mellitus, previous ischemic stroke or TIA, myocardial infarction, heart valve disease, smoking and peripheral artery disease). Hypertension was defined as a systolic blood pressure > 140 mmHg or a diastolic blood pressure > 90 mmHg. Diabetes mellitus or hypercholesterolemia were defined as present if named in medical history. Furthermore, to assess differences in functional independence prior to stroke the modified Rankin Scale (mRS) was collected, also called the pre-mRS. The national Institute of Health Stroke Scale (NIHSS) was collected to assess stroke severity at admission.

### 2.3. Outcomes

All laboratory blood tests were done within 24 h after stroke onset. For the purpose of this sub study leukocyte and thrombocyte levels were assessed. The primary outcome measure of this sub study was the presence of ICAS (defined as a stenosis ≥ 50%) in relation to the level of thrombocytes and leukocytes. The percentage of stenosis was assessed on either computed tomography angiography (CTA) or on carotid ultrasound examination. For measuring degree of stenosis on CTA the NASCET criteria were used, if carotid ultrasound was performed a standardised ratio to determine percentage of stenosis was used, as recommended by the radiologists association.

### 2.4. Statistical Analysis

Baseline characteristics of male and female patients are compared. Presence of ICAS was compared with leukocyte and thrombocyte serum levels. Intergroup comparisons were performed using chi-square test, independent samples *t*-test, or Mann–Whitney U test, as appropriate. No imputation was used for descriptive variables and the number of missing values is reported. For the primary outcome, groups were stratified by sex. Logistic regression was performed to assess the relationship of leukocyte and thrombocyte serum levels and the presence of ICAS. The primary outcome was adjusted for age, race, systolic blood pressure, diastolic blood pressure, medical history (stroke, hypercholesterolemia, hypertension, myocardial infarction, cardiac valve disease, peripheral vascular disease), smoking, baseline NIHSS, antiplatelet therapy, anticoagulant use and use of statins. Common odds ratio’s with 95% confidence interval (CI) and their corresponding *p*-values were calculated for leukocytes and thrombocytes. All variables used in multivariable logistic regression were more than 98% complete, so simple imputation by the mode was used for categorical variables and single imputation by the mean for continuous variables. Statistical analyses were performed using SPSS (Version 26, IBM, Armonk, NY, USA). A *p*-value equal or less than 0.05 was deemed significant.

## 3. Results

Between 6 July 2010, and 23 March 2014, a total of 2550 patients were randomly assigned to the two treatment groups in PASS. During the PASS, clinical data about the presence and the percentage of ICAS was collected for 1480 patients, including 1455 (98%) with TIA or an ischemic stroke (Figure 1). Subsequently, nine (0.6%) had hyperleukocytosis (defined as leukocyte count ≥ 50 × 10^9^/L) and 33 (2%) had both atrial fibrillation and ICAS, leaving 1413 patients for the current analysis.

The median age of evaluated patients was 73 years [IQR, 64–82] and 577 (41%) were female. Female patients were older (median age 75 vs. 71, *p* < 0.001) and presented with lower diastolic blood pressure (84 mmHg vs. 89, *p* < 0.001) than male patients. The proportion of patients with a medical history of hypertension was higher among females than males (59% vs. 51%, *p* = 0.005). Less female patients had suffered from myocardial infarction (7% vs. 16%). Use of antiplatelet therapy (37 vs. 44%, *p* = 0.018) and statins (33% vs. 42%, *p* < 0.001) prior to presentation was less frequently observed in female patients than in male patients. Current smoking did not differ significantly between sexes. See Table 1 for all baseline characteristics.

An ipsilateral carotid artery stenosis (≥50%) was less frequently observed in female patients than in male patients (12% vs. 20%, *p* < 0.001). Female patients with ICAS showed a mean for leukocytes of 9.31 × 10^9^/L and 273 × 10^9^/L for thrombocytes compared to a mean of 8.44 × 10^9^/L for leukocytes and 256 × 10^9^/L for thrombocytes in female patients that displayed no ICAS. Male patients had a mean of 9 × 10^9^/L for leukocytes and 234 for thrombocytes when ICAS was present, while this was a mean of 8.1 × 10^9^/L for leukocytes and 222 × 10^9^/L for thrombocytes in male patients where ICAS was not found. Thrombocyte count was higher in female patients than in male patients (259 × 10^9^/L vs. 224, *p* < 0.001). Leukocyte count did not differ significantly between male and female patients. The pre-mRS was significantly higher for female patients than male patients, albeit showing the same median and interquartile ranges (1 (1–2) vs. 1 (1–2), *p* = 0.001). NIHSS scores at presentation were higher in female patients compared to male patients (5 (3–9) vs. 4 (3–7), *p* = 0.006)).

Leukocytes were significantly higher in both female and male patients with ICAS patients compared with non-ICAS patients. No difference was seen between the groups looking at thrombocytes. Serum levels of leukocytes, CRP and thrombocytes comparing ICAS patients with non-ICAS patients per gender are shown in Table 2.

In univariable logistic regression analysis, leukocytes were predictive for the presence of ICAS in both female and male patients. When adjusted, leukocytes remained a significant predictor only in male patients, but not in female patients (see Table 3). Interaction *p*-values were all >0.05. Thrombocyte count was not a significant predictor of ICAS in both female and male patients. Excluding patients with TIA’s did not alter these associations.

## 4. Discussion

This study aimed to assess if leukocytes and/or thrombocytes are associated with presence of ICAS ≥50% in the acute stroke setting. We showed that leukocytes, but not thrombocytes are associated with the presence of ICAS ≥50% in male patients in the acute stage after ischemic stroke or TIA, but not in women. To our understanding this is the first study that showed this association in the acute moment after ischemic stroke (<24 h after onset).

Strength of this sub study is the large number of patients with 1413 patients used in the final analyses. Another strength of this study is that the analyses were all corrected for the known predictors of ICAS in order to minimalize any confounding. Due to necrosis of brain tissue, the association between elevation of leukocytes post-ischemic stroke and ICAS could be confounded by the size of the ischemic area. As the best derivative of stroke size, NIHSS was used as variable to correct for this possible rapid rise in leukocyte number.

Our finding that leukocytes are associated with the presence of ICAS in men is in line with the results of the Tromsø study. In this study, consisting of 5341 persons (2982 woman and 2359 men), leukocytes were shown as a predictor of presence of ICAS [6]. In this population-based trial, the population was largely asymptomatic, with 4% of women versus 5% of men which had an ischemic stroke before the start of the study. The main aim of this study was to determine any differences between women and men and to identify any predictors for plaque echogenicity.

Earlier studies looking at the systematic inflammatory response following a stroke, found that circulating levels of inflammatory cytokines increase in response to ischemia [10]. Furthermore Interleukin(IL)-6 levels are positively correlated with stroke severity [11]. However, after adjustment for NIHSS, an indirect measure of stroke severity, leukocytes remained a significantly associated with ICAS. The results of our proof of principle sub study further demonstrate that, when adjusted for other variables than NIHSS, namely cardiovascular diseases, age and smoking, leukocyte count is a viable independent predictor for the presence of ICAS in men. Smoking, and age are known to alter levels of serum inflammatory cytokines and leukocytes [12,13]. However, adjustment for these factors did not alter the significant association of leukocytes in predicting presence of ICAS. These results could suggest a role for inflammation that is more important and potentially more substantial than thought before.

A limitation of this sub study was that of the 2550 patients of the PASS study, data about the internal carotid artery was collected for 1480 patients and for only 992 patients CRP levels were recorded, making the data prone to selection bias. This is partly caused by the inclusion of patients with haemorrhagic stroke in PASS, as in these patients no information regarding ICAS is obtained. Additionally, a specific form of imaging was not required to assess stenosis grade of the internal carotid artery. In some patients ultrasound was performed in others CTA of the carotid artery. In a future study stenosis grade should be assessed in all patients with the same form of imaging. Preferably, at least ultrasound should be performed and echogenicity of the plaque should be assessed in all patients.

Another limitation is the use of NIHSS to correct for stroke severity in this sub study. Severity of stroke symptoms could correlate with actual stroke volume and actual stroke volume could correlate with blood leukocyte count. You should therefore correct for actual stroke volume. Preferably, an MRI or CT of the brain to investigate actual stroke volume should be performed. This was not done in PASS, but should be part of a future study investigating the role of inflammation in the acute stroke setting.

Blood collection was done in the 24 h following onset of neurological symptoms, but exact time of blood collection was not specified. Although not likely, differences in haematological and inflammatory markers may therefore be explained by temporal differences in immune response following ischemic stroke in all patients and thus not have a predictive value for presence of ICAS [10,11,14]. Furthermore, some variables that may interact with leukocyte level, such as adiposity, where not collected and therefore adjustment for these variables could not be done. There could be potential confounding due to adiposity and other unknown confounders that are known to have a causative factor in atherosclerosis and ICAS.

This study is one of the first studies that looked at haematological and inflammation markers in relation to ICAS during acute stroke (<24 h after onset of symptoms). In the setting of carotid endarterectomy (CEA) for ischemic stroke caused by ICAS, previous studies looked at serum markers, but most of these studies had a small number of patients [15,16,17,18]. Compared to these studies, our study was not constricted by a small sample size. This study can be a step in finding reliable predictors for high risk ICAS in the acute stroke setting.

Several other studies looked at specific white blood cells instead of leukocytes as a group, in relation to cardiovascular diseases. To be able to use the different subsets of leukocytes, a leukocyte differential count should be performed. One study focussed on coronary artery disease risk and found that both neutrophils and lymphocytes predicted a high risk for a cardiovascular event, but the best prediction could be given by the neutrophil to lymphocyte ratio [19]. If any ratio using different leukocytes, as assed by leukocyte differential count, have a predictive value in ICAS remains enigmatic. It does, however, underline the importance of a leukocyte differential count being part of the future prospective study described above.

The precise role of inflammatory markers in atherosclerosis remains unknown [1]. The elevated leukocytes found in our study can either be a true atherosclerosis risk factor or it reflects the inflammatory activity of the carotid plaque [1]. The increased inflammatory activity could be caused by atherosclerotic lesions or it could be the cause of the atherosclerotic lesions. Another hypothesis is that the elevated leukocyte levels are caused by release of cytokines due to ulceration of the carotid plaque and the subsequent release of plaque content into the bloodstream. Future research should distinguish the role of serum inflammatory makers and identify which of these hypotheses is true.

In Colchicine for Prevention of Vascular Inflammation in Non-cardio Embolic Stroke (CONVINCE) randomized clinical trial, patients that had a non-cardiac ischemic stroke are given colchicine combined with standard stroke care or standard stroke care alone [20]. This study investigates whether treatment with immune altering medication can have an effect on stroke recurrence. This trial may provide some evidence whether systemic inflammatory markers have an active role in atherosclerosis by looking at recurrence rates in patients with a symptomatic stenosis within this trail and if treatment can reduce recurrent events. And if so, if leukocyte count is a marker for treatment effect.

Other research that can indicate if serum levels reflect the activity within the atherosclerotic plaque would be to investigate promising inflammatory biomarkers, such as TNF-α, IL-1β, IL-18, white blood cell levels and more specifically monocytes in relation to vulnerable plaque characteristics, as seen on MRI. For instance, in carotid atherosclerotic plaque of patients undergoing CEA, histopathological analysis showed an upregulation, IL-1β and IL-18, in which higher levels were found in vulnerable plaques than in stable plaques [21]. Patient outcomes should also be assessed in these studies, such as stroke recurrence. In this way, these future studies can possible provide us with reliable serum marker predictors and help in optimizing risk stratification of ICAS patients in the acute setting or assist in therapy choice.

This was the one of the first cross-sectional studies examining the association between leukocyte count and presence of internal carotid artery stenosis ≥50% in the acute stroke setting. We found an independent association between leukocyte count in men, but not in women. No association with thrombocyte count was found. Further research linking promising inflammatory serum markers with imaging characteristics of plaques and patient outcome is warranted, specifically with a prospective study design, differential leukocyte count, assessment of echogenicity on ultrasound and with brain imaging to assess stroke volume. Next to additional exploration of the precise role for leukocytes in the acute stroke setting, eventually through a large study collecting blood on a protocolled manner to minimize bias and examine the increase of leukocyte count over time following ischemic stroke.

## Figures and Tables

**Figure 1 jcm-11-07286-f001:**
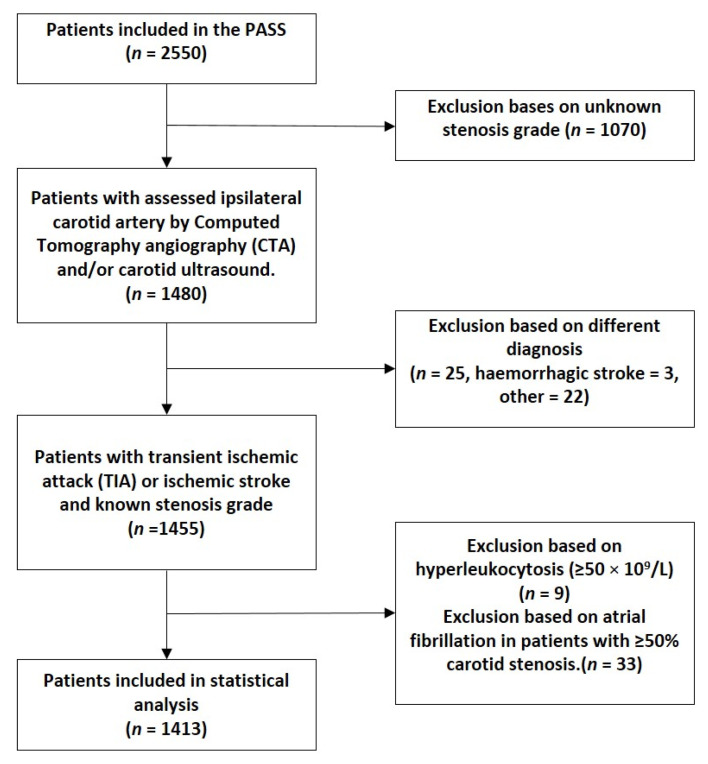
Flowchart of included patients.

**Table 1 jcm-11-07286-t001:** Baseline characteristics.

	Sex		
	Female (*n* = 577)	Male (*n* = 836)	*p*-Value
Age, years	75 (65–82)	71 (62–79)	<0.001
Caucasian	540/577 (94%)	782 (94%)	<0.972
Systolic blood pressure (mmHg)	165 (162–167)	164 (162–166)	0.670
Diastolic blood pressure (mmHg)	84 (83–86)	89 (87–90)	<0.001
Carotid artery stenosis (≥50%)	73/577 (13%)	166/836 (20%)	<0.001
Serum markers			
Leukocytes (×10^9^/L)	8.55 (8.29–8.75)	8.29 (8.1–8.46)	0.077
Thrombocytes (×10^9^/L)	259 (252–265)	224 (219–229)	<0.001
Medical History			
Atrial fibrillation/Flutter	62/575 (11%)	84/834 (10%)	0.667
Stroke	177/575 (31%)	272/836 (32%)	0.487
Hypercholesterolemia	141/572 (25%)	240/827 (29%)	0.071
Hypertension	338/576 (59%)	427/836 (51%)	0.005
Myocardial infarction	39/575 (7%)	137/836 (16%)	<0.001
Cardiac valve disease ^a^	33/576 (6%)	46/834 (6%)	0.864
Peripheral vascular disease	39/574 (7%)	71/834 (9%)	0.237
Diabetes Mellitus	103/576 (18%)	166/836 (20%)	0.353
Malignancy	49/576 (9%)	61/834 (7%)	0.411
Current smoker	141/573 (25%)	239/834 (29%)	0.093
Previous medication			
Anticoagulants	41/577 (7%)	63/836 (8%)	0.761
Antiplatelet therapy	216/577 (37%)	365/835 (44%)	0.018
Statins	188/577 (33%)	347/835 (42%)	<0.001
Modified Rankin Scale before stroke symptoms (pre-mRS) ^b^	1 (1–2)	1 (1–2)	0.001
Baseline National Institutes of Health Stroke Scale (NIHSS) ^b^	5 (3–9)	4 (3–7)	0.006
Treatment			
Intravenous thrombolysis	220/577 (38%)	325/836 (39%)	0.777
Post-stroke Ceftriaxone	289/577 (50%)	421/836 (50%)	0.920

Data are median (IQR), mean (95% CI) or *n*/N (%). ^a^ Cardiac valve disease was defined as cardiac valve insufficiency, stenosis, or replacement. ^b^ The mRS is a scale ranging from 0 to 6, with 0 being absence of symptoms and 6 indicating death; the mRS before onset of symptoms was assessed in all 1413 patients. The NIHSS ranges from 0 to 42, with a higher number indicating a higher degree of stroke severity.

**Table 2 jcm-11-07286-t002:** Serum markers in patients with and without internal carotid artery stenosis.

	Internal Carotid Artery Stenosis (<50%)	Internal Carotid Artery Stenosis (≥50%)	*p*-Value
Female patients	*n* = 507/577 (87%)	*n* = 70/577 (13%)	
Leukocytes (×10^9^/L)	8.44 (8.21–8.67)	9.31 (8.51–10.1)	0.041
Thrombocytes (×10^9^/L)	256 (250–263)	273 (252–295)	0.128
CRP (×mg/L)	8.25 (6.51–9.99)	11.68 (3.26–20.10)	
Male patients	*n* = 670/836 (80%)	*n* = 166/836 (20%)	
Leukocytes (×10^9^/L)	8.1 (7.91–8.3)	9 (8.58–9.17)	<0.001
Thrombocytes (×10^9^/L)	222 (216–227)	234 (223–244)	0.064
CRP (×mg/L)	8.09 (6.63–9.54)	8.78 (6.22–11.35)	

Data mean (95% CI) or *n*/N (%).

**Table 3 jcm-11-07286-t003:** Leukocytes and thrombocytes as predictors of the presence of ICAS.

	Unadjusted Model		Adjusted Model	
	Common Odds Ratio	*p*-Value	Common Odds Ratio	*p*-Value
Female patients (*n* = 577)				
Leukocytes (×10^9^/L)	1.098 (1.017–1.186)	0.017	1.041 (0.995–1.136)	0.360
Thrombocytes (×10^9^/L)	1.002 (0.999–1.005)	0.130	1.002 (0.999–1.005)	0.145
Male patients (*n* = 836)				
Leukocytes (×10^9^/L)	1.131 (1.064–1.202)	<0.001	1.094 (1.024–1.169)	0.008
Thrombocytes (×10^9^/L)	1.002 (1.000–1.404)	0.065	1.002 (0.999–1.004)	0.160

Adjusted model is adjusted for age, race, current smoking, systolic blood pressure, diastolic blood pressure, hypertension, hypercholesterolemia, previous stroke, diabetes mellitus, heart valve disease, peripheral vascular disease, myocardial infarction, malignancies, treatment with anticoagulants, antiplatelet therapy and statins and NIHSS at presentation.

## Data Availability

The datasets used and/or analysed during the current study are available from the corresponding author on reasonable request.

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
