# Peer review of "Leukocyte Count Predicts Carotid Artery Stenosis in Men with Ischemic Stroke: Sub Study of the Preventive Antibiotics in Stroke Study (PASS)"

_jcm, 2022, doi:10.3390/jcm11247286_

Round 1
Reviewer 1 Report
The present study is a subanalysis of the authors' prospective clinical study, PASS. Based on the results of a known study (Tromsø study) showing an independent association between blood leukocyte count and carotid plaque area in men, but not in women, the authors examined the sex-dependent association between leukocyte count and carotid artery stenosis in the acute stroke period. The results showed that leukocytes were independently associated with ICAS >50% in men, but not in women.
1. The novelty of the present study is that the results in the acute phase of stroke were similar to those of the Tromsø study. However, it has already been reported in the Tromsø study that plaque echogenicity, stenosis, and leukocyte count were independent predictors of cerebrovascular event occurrence, and the present study confirms this finding. Therefore, the originality of the present study does not appear to be very high.
2. In the present statistical analysis, NIHSS was used to correct for cerebral infarct size. However, while NIHSS values correlate with cerebral infarct volume, they are not necessarily identical. Measuring its volume from images (CT and/or MRI) at the time of stroke onset is considered more accurate.
3. The Tromsø study reported that WBC was not correlated with plaque area in women, but was significantly associated with plaque echogenicity. Was there a correlation in this regard in the present study?
Reviewer 2 Report
The study investigated the role of leukocytes and thrombocytes in carotid artery stenosis in patients with ischemic stroke. The study found in patients with ischemic stroke, the leukocyte is associated with carotid artery stenosis in men but not in women. Thrombocyte is not associated with carotid artery stenosis. The study still has some issues that should be clarified.
1. In the study, some patients received a carotid ultrasound or angiography study for evaluating the condition of carotid stenosis but some did not. The author should state the criteria for performing a carotid ultrasound or computed tomography angiography study.
2. The methods for evaluating stenosis rate should be stated.
3. Page 5, In lines 133 to 134, Serum levels of leukocytes, CRP, and thrombocytes
comparing 133 ICAS patients with non-ICAS patients per gender are shown in
table 2. But in table 2, we cannot find data about the level of CRP.
Round 2
Reviewer 1 Report
The authors presented a minor revision of the paper, but some changes and additions are needed in the introduction and discussion sections.
1) In the Abstract, the authors should state what PASS stands for and provide a brief description of the study in the Background section. In the Methods section, "In PASS patients with ischemic stroke or TIA were randomized between four days of ceftriaxone intravenously or standard stroke care alone.” seems unnecessary.
2) A brief description of PASS (e.g., what the purpose of the study is, what kind of study it is, etc.) should be given in the introduction.
3) The authors should include the information described in “Replies for reviewer 1” #2. and #3. as limitations in the discussion.
4) There are some typographical errors.
Author Response
Dear editor and reviewer,
Thanks again for your comments on our manuscript. We implemented the comments in the manuscript as well as the previous comments. In this text we describe what we did with your comments:
1) We adjusted the abstract according to your comment.
2) A short introduction to PASS is given in the introduction. We decided to keep it short, since the methods section starts with a more elaborate description of PASS.
3) Two paragraphs were added as limitations in the discussion section, with the same information that was given in the reply to the previous comments.
4) The whole text was screened for typographical errors and they were corrected.
If any comments or questions remain, please let us know through email! Thanks a lot for the quick report!
Kind regard,
On behalf of all the authors,
Twan van Velzen